# Assessing Phenotypic and Genotypic Resistance to Flumethrin in *Varroa destructor* Populations in Muğla, Türkiye

**DOI:** 10.3390/insects16060548

**Published:** 2025-05-22

**Authors:** Ali Sorucu, Bekir Çöl, Esra Dibek, Anara Babayeva

**Affiliations:** 1Department of Pharmacology and Toxicology, Faculty of Milas Veterinary Medicine, Muğla Sitki Koçman University, Muğla 48200, Türkiye; 2Beekeeping and Silkworm Research and Application Centre, Muğla Sıtkı Koçman University, Muğla 48200, Türkiye; 3Department of Biology, Faculty of Science, Muğla Sitki Koçman University, Muğla 48100, Türkiye; 4Biotechnology Research Centre, Muğla Sitki Koçman University, Muğla 48100, Türkiye; esradibek@mu.edu.tr (E.D.); anara.babazade@gmail.com (A.B.); 5Department of Pharmacy Services, Köyceğiz Vocational School of Health Services, Muğla Sitki Koçman University, Muğla 48800, Türkiye; 6Biology Program, Graduate School of Natural and Applied Sciences, Muğla Sitki Koçman University, Muğla 48000, Türkiye

**Keywords:** *Varroa destructor*, resistance, flumethrin, phenotypic, genotypic

## Abstract

Varroa destructor (varroa) is a major parasitic threat causing significant colony losses around the world. While flumethrin remains a widely used acaricide, its efficacy has declined substantially due to evolving pesticide resistance, exacerbating colony and yield losses. This study evaluates flumethrin resistance through LD50 analysis and mutational profiling of the voltage-gated sodium channel (VGSC) L925 locus. Our findings reveal dramatically elevated LD50 values compared to previous studies, indicating widespread resistance. Notably, 95% of the population carried homozygous resistant alleles (L925I/M/V), underscoring the urgent need for alternative control strategies.

## 1. Introduction

*Varroa destructor* is one of the most significant threats to honeybee health due to the direct damage it inflicts, the physiological stress it induces, and its role as a vector for secondary pathogens [1]. Therefore, effective varroa management is critical to sustaining global apiculture [1]. Beekeepers employ diverse strategies to control varroa mites, including physical, biological, chemical, and genetic methods. Chemical interventions, such as synthetic acaricides (e.g., amitraz, coumaphos, tau-fluvalinate, and flumethrin), organic acids, and essential oils, remain central to these efforts [2]. The choice of treatment depends on environmental and hive-specific factors, including temperature, humidity, brood presence, and proximity to foraging seasons [2]. However, physical and biological methods often prove insufficient, necessitating integration with chemical controls to achieve adequate mite suppression [3].

Among synthetic acaricides, pyrethroids like flumethrin are widely used due to their neurotoxic mechanism: they disrupt insect nerve function by prolonging sodium channel activation during neural signalling, leading to spastic paralysis and death [4,5]. For example, commercial plastic strips impregnated with 3.6 mg of flumethrin are applied in hives to eliminate varroa via contact exposure [4,5]. Despite their efficacy, overreliance on pyrethroids has driven the evolution of resistance in varroa populations, resulting in treatment failures and escalating acaricide residues from higher dosage applications [6,7,8]. Some studies clearly show that the effectiveness of pyrethroids to varroa is reduced [9,10]. Flumethrin resistance was first documented in Italy and has since spread across Europe (e.g., Slovenia, Switzerland, Poland, France, Belgium, England, Austria) and beyond, with confirmed cases in Mexico and Uruguay [6,7,8]. Phenotypic resistance is increasingly evident in elevated LD_50_ (median lethal dose) values reported globally, though no prior studies have quantified flumethrin-specific LD_50_ levels in Türkiye [4,11,12,13].

In Türkiye, molecular evidence of resistance has emerged in studies which detected mutations in the voltage-gated sodium channel (VGSC) gene (L925 locus) linked to pyrethroid resistance in 69–75% of varroa samples from some regions in Türkiye [14,15,16]. However, in Turkish varroa populations, phenotypic resistance levels (e.g., LD_50_) and their correlation with genotypic mutations remain uncharacterized.

This study investigates both genotypic and phenotypic resistance to flumethrin in *Varroa destructor* from Muğla, a region pivotal to Turkish beekeeping due to its high colony density, pine honey production, and role as a wintering hub. We quantified acute LD_50_ values to assess phenotypic resistance and analyzed the VGSC L925 locus for mutations associated with pyrethroid resistance. Our findings may inform adaptive management strategies to mitigate resistance-driven colony losses.

## 2. Materials and Methods

### 2.1. Collection of Varroa Samples

The varroa mites used in this study were collected from 27 apiaries in the districts of Milas, Dalaman, Menteşe, Köyceğiz, and Ula, which are in the Muğla region of Türkiye, from May to October 2021 (Table 1). Two collection methods were used in the preliminary trials: collection from brood cells and powdered sugar on adult bees. The powdered sugar method was selected due to higher mite yields (Figure 1). Beekeepers were recruited through the Muğla Beekeepers Association, Provincial and District Directorates of Agriculture and Forestry. In preliminary interviews with the beekeepers, apiaries that used flumethrin at least once a year were preferred. A3 paper was placed on the hive floor or pollen drawer, and powdered sugar was sprayed between the frames to collect varroa from the hives.

The aim was to collect 100 to 150 live varroa from each apiary. The collection paper on the hive floor was carefully removed 5 min after applying the powdered sugar. A paintbrush was used to separate powdered sugar residues from the fallen varroa mites (Figure 1). To ensure that the collected mites had survived the collection and cleaning procedures, they were held in petri dishes (an average of 100–150 varroa per apiaries) with a small paper towel inside. Only the varroa mites that remained alive after 3 h post-collection procedures were used in the LD_50_ test (Figure 2). The varroa mites that moved after the paintbrush was touched and attached to the paintbrush were considered alive and separated for the LD_50_ study. After completing the LD_50_ tests, the varroa mites were stored at −20 °C for further analyses of flumethrin resistance mutations.

Varroa mites were exposed to flumethrin in a single Petri dish per dose (no biological replicates). Ten mites were tested per dose, and mortality was recorded. The term “Average Mortality” in Table 1 refers to pooled data across three apiaries, not technical replicates.

### 2.2. Acute LD_50_ Analysis and Determination of Phenotypic Resistance

The lethal dose (LD_50_) study was conducted by modifying the method of Maggi et al., 2008 [12]. Varroa samples were collected from 3 different apiaries, and susceptibility ranges were determined by preliminary testing. Bayticol^®^ (flumethrin 1% Bayer-Germany) was used to prepare different concentrations of flumethrin (2 mL). Dilutions of flumethrin were made using hexane. The dilutions of flumethrin were loaded in 13 petri dishes with 10,000 µg, 7500 µg, 5000 µg, 3600 µg (amount in the original preparation), 2500 µg, 1000 µg, 500 µg, 250 µg, 100 µg, 50 µg, 25 µg, and 10 µg flumethrin and only hexane as a control. To ensure homogeneous distribution of flumethrin on the petri dishes, a further 2 mL of hexane was added to each petri dish after application, spread well on the bottom, and then dried at room temperature. In preliminary studies, the average mortality ranged from 50 µg to 25 µg, and all varroa died at 100 µg. As a result of the preliminary experiment, petri dishes were prepared by loading 0, 10, 20, 40, 50, 60, 60, 80, and 100 µg of flumethrin and a control (hexane). After assessing varroa mite viability, 8–13 varroa mites were placed in a single Petri dish per dose (no biological replicates), exposed to flumethrin for two hours, and then reassessed for viability by touching a paintbrush (Figure 2). The number of varroa placed in the petri dishes decreased to 8 in locations where live varroa mites were low and increased up to 13 in locations where live varroa mites were high. A separate paintbrush was utilized for each dosage to prevent contamination in the control. Dead and live varroa mites in petri dishes were recorded for LD_50_ analysis. The averages were obtained by optimizing the mortality according to 10 varroa mites, as shown in Table 1. The calculated LD_50_ value was written over this optimized value.

Acute median lethal dose (LD_50_) analysis was conducted using the Karber–Behrens method, a simplified approach providing a rough estimate without confidence intervals or statistical modelling of dose–response curves with the formula LD_50_ = LD100 − [Σ(ab)/n]. LD100 is the dose that caused 100% mortality in the experiment, n is the number of animals in each group, a is the difference between two consecutive doses, and b is the arithmetic means of the deaths caused by two consecutive doses. While logistic regression would offer greater rigor, resource constraints precluded its use. These limitations should be considered when interpreting LD_50_ reliability.

Finally, the varroa mites were stored in Eppendorf tubes at −20 °C for further analyses of flumethrin resistance mutations.

### 2.3. DNA Isolation from Varroa Samples

High-quality DNA could not be isolated from 5 apiaries (V23–V27) out of the 27 sampled, so molecular analyses proceeded with 22 apiaries. First, varroa samples were sorted by location and labelled (V1–V22). For each location, five mites were pooled into a single tube, and DNA was isolated using a commercial kit (GeneAll Exgene Kit, Animal Tissue protocol, GeneAll Biotechnology, Seoul, South Korea).

Due to the tough exoskeleton of varroa mites, the samples were mechanically disrupted using glass beads of varying sizes and lysis buffer, followed by vigorous vortexing. Next, 250 µL of lysis buffer (BL) was added, and the samples were vortexed for 5 min, incubated at 70 °C for 10 min, vortexed again, and flash-frozen at −80 °C for 10 min. This freeze–thaw–vortex cycle was repeated three times, as per the kit protocol. DNA was eluted twice with 40 µL of AE buffer, quantified via Nanodrop spectrophotometry, and visualized on 1% agarose gel for quality assessment. Isolated DNA was stored at −20 °C.

### 2.4. Amplification of the VGSC Gene Region by PCR Using Appropriate Primers

Due to limited DNA quantities, 5 µL of each sample was used for preliminary PCR screening with RAPD primers (OPB-05) to confirm amplifiable DNA. The reaction mixture included template DNA, 10 µM OPB-05 primer, 200 µM dNTP mix, 1 U Taq DNA polymerase, 1× PCR buffer, 1.5 mM MgCl_2_, and nuclease-free water. Successful amplification with RAPD primers confirmed DNA integrity. Subsequently, the voltage-gated sodium channel (VGSC) gene region was targeted using primers 1273iF (5′-AAGCCGCCATTGTTACCAGA-3′) and 1973iR (5′-GCTGTTGTTACCGTGGAGCA-3′), validated by Millán-Leiva et al. (2018) [17]. Reactions were performed in triplicate under identical conditions: initial denaturation: 94 °C for 2 min, 35 cycles of denaturation: 94 °C for 30 s, annealing: 51 °C for 30 s, extension: 72 °C for 60 s, and lastly, additional final extension: 72 °C for 10 min.

PCR products were resolved on 0.8% agarose gels, stained with ethidium bromide, and visualized under UV light.

### 2.5. RFLP Analysis of the PCR Products

Restriction Fragment Length Polymorphism (RFLP) analysis was performed on the amplified PCR products, following the protocol of Millan-Leiva et al. (2018) [17]. Briefly, PCR products were digested with the SacI restriction enzyme (Thermo Fisher Scientific, Waltham, MA, USA). The resulting restriction fragments were then separated by electrophoresis on a 2% TBE agarose gel. Following electrophoresis, the gel was stained, visualized under UV light, and the banding patterns were documented using a gel imaging system.

### 2.6. DNA Sequencing of Some of the VGSC PCR Products by Sanger and Bioinformatics Analysis

Based on RFLP results of the varroa samples from 22 locations, 12 samples (V1, V2, V4−V10, V20, V22) were selected for Sanger sequencing to confirm the absence of homozygous susceptible (SS) genotypes, verify the *Sac*I restriction enzyme digestion efficiency, and resolve codon-level genotypic ambiguities. This enabled the simultaneous validation of RFLP interpretations and the detection of novel mutations. Chromatograms were analyzed using BioEdit v7.2.6 and MEGA X for base calling, alignment, and phylogenetic analysis, with sequence homology confirmed via BLASTN (NCBI Nucleotide Database, 2024, The National Center for Biotechnology Information, Bethesda, MD, USA).

## 3. Results

The mortality of the varroa samples by dose is presented in Table 1. No mortality was observed in the control group. LD_50_ values ranged from 31 µg (lowest) to 61.8 µg (highest), averaging 49.13 µg, with most values clustering between 40 and 50 µg (Table 1).

DNA isolation from 22 varroa samples (collected from 27 beekeepers) was performed following the protocol. Agarose gel electrophoresis and Nanodrop measurements confirmed successful isolation. PCR amplification of the VGSC gene region was achieved for all 22 locations.

### 3.1. RFLP Analysis of PCR Products

The PCR products were analyzed via Restriction Fragment Length Polymorphism (RFLP) using the *Sac*I restriction endonuclease enzyme. Digested fragments were separated on 2% TBE agarose gel (Figure 3). No bands corresponding to susceptible (SS) genotypes (438 bp and 263 bp) were observed, confirming widespread resistance. Allele frequencies were quantified using ImageJ and visual inspection [18], with the results categorized by genotype (Table 2).

RFLP band densities were examined to detect allele frequencies in the mites using the approach of Stara et al. (2019) using the ImageJ software (version 1.54h, The National Center for Biotechnology Information, Bethesda, MD, USA) and visual observation [18]. When interpreting the PCR-RFLP bands after SacI enzyme cleavage, only two fragments (bands) of 438 and 263 bp should have been observed due to the presence of the codon encoding the amino acid leucine (CTG) in the drug-sensitive homozygous (SS) mites. This was not detected in the samples when the gel images were analyzed. According to the RFLP results of the samples, the VGSC (voltage-gated sodium channel) profiles were evaluated and separated according to allele groups (Table 2).

### 3.2. Bioinformatics Analysis of VGSC PCR Products

Sanger sequencing chromatograms from 12 locations (V1, V2, V4–V10, V20, V22) were analyzed using the BioEdit software version 7.2.5 (see Mitton et al., 2021) [19]. Sequence results were analyzed using Bioedit, MEGA X, and BlastN to identify mutations (alleles) [19]. DNA isolation and PCR were performed on five mites per location. Genotypes were determined by integrating PCR-RFLP band intensities, as described by Stara et al., 2019 [18], and sequence chromatogram profiles (as reported by Mitton et al., 2021) [19]. The resulting genotypes of 110 Varroa mites are summarized in Table 3.

Sequence analysis confirmed the Leu (CTG) → Ile (ATA) mutation in V1 (V1a–V1e), V2 (V2a–V2e), V4 (V4a–V4e), V5 (V5a–V5e), V6 (V6a–V6e), V7 (V7a–V7e), and V22 (V22a–V22e) (Figure 4, Appendix A). The combined RFLP and sequencing results classified all five individuals per sample as homozygous resistant (Ile/Ile) (Table 3).

Sequence analysis of the V8 sample revealed the Leu (CTG) → Ile (ATA) mutation (Figure 5). When the genotype was interpreted according to the RLFP and sequence results, in general, 70% of the alleles were resistant alleles (Ile/Ile), and 30% were GTG (Val) alleles (Table 3). Sequence analysis of the V9 sample clearly showed the Leu (CTG) → Ile (ATA) mutation (Appendix A). When the genotype was interpreted according to the RLFP and sequence results, in general, 70% of the alleles were resistant alleles (Ile/Ile), and 30% were ATG (Met) alleles (Table 3).

Analysis of the sequence of the V10 and V20 samples showed the Leu (CTG) → Ile (ATA) mutation (Appendix A). When the genotype was interpreted according to RLFP and sequence results, 70% of the alleles generally contained resistant alleles (Ile/Ile), and 20% contained ATG (Met) alleles. In addition, 10% of the alleles contained the CTG (Leu) S allele (Table 3).

In addition, mutations were detected in different regions of the VGSC, i.e., regions 147: TGTCGA → TGTCAA of the V9 samples (Appendix A); region 149: GTCGAG → GTCAAG of the V5 samples (Appendix A); region 150: TGTCGA → TGTCAA of the V7 and V8 samples (Figure 5 and Appendix A); and region 423: CAGGGA → CGGGGA of the V1 samples (Appendix A). While these mutations are not currently linked to resistance, their functional roles warrant further investigation.

A total of 110 mites were analyzed (22 pooled samples × 5 mites per pool), yielding 220 alleles (diploid genome). RFLP analysis identified RR (homozygous resistant) and SR (heterozygous) genotypes in the pooled samples. Allele frequency analysis revealed 103 resistant (R) alleles encoding isoleucine (Ile, ATA), 7 methionine (Met, ATG), and 3 valine (Val, GTG). RFLP data corroborated 96 R alleles, with 209/220 alleles (95%) classified as resistant. Only 11 susceptible (S) alleles (Leu, CTG) were detected, indicating that the heterozygous pools retained residual susceptibility.

Isoleucine (Ile) predominated among the resistant alleles (91%), followed by methionine (6%) and valine (3%). With S alleles constituting just 5% (11/220) of the population, directional selection will likely drive near-complete fixation of R alleles in subsequent generations. Consequently, 95% of the mites were homozygous resistant (RR), while 5% were heterozygous (SR).

## 4. Discussion

In the present study, varroa mite samples were exposed to different concentrations of flumethrin acaricide, and acute LD_50_ values were calculated. However, since all the individuals subjected to molecular analysis were resistant, comparisons of LD_50_ values between resistant and susceptible populations were unattainable. For meaningful interpretation of LD_50_ data, baseline susceptibility levels must first be established by characterizing susceptible individuals within the population. Additionally, while three distinct mutations were identified in the gene encoding the voltage-gated sodium channel (VGSC) protein—the target site of flumethrin—the relationship between these mutations and dose-dependent mortality warrants further investigation. Flumethrin is typically introduced to hives via impregnated strips (commonly 3.6 mg), where it translocates into bee fat or tissues. Varroa mites succumb to the acaricide through direct contact or ingestion during bee feeding. The calculated average LD_50_ of 49.1 µg indicates the dose required to achieve 50% mortality in the tested population. Consistent with findings by Wu et al. (2023), we propose applying varying flumethrin concentrations to honeybees, quantifying residue levels in bee tissues, and correlating these with varroa LD_50_ thresholds to refine field dosing strategies [20].

Furthermore, the toxicity of these doses to honeybees and their behavioural effects warrant investigation. Previous studies employing similar methodologies reported significantly lower LD_50_ values: Maggi et al. (2008), in Argentina, documented an LD_50_ of 0.34 µg, Goodwin et al. (2005), in New Zealand, observed 12 µg, and Mitton et al. (2016), in Uruguay, identified 3.8 µg [4,12,13]. However, a recent study by McGruddy et al. (2024) in New Zealand reported an LD_50_ of 156 µg—3-fold higher than our findings (49.1 µg) and 26-fold higher than earlier New Zealand data [11,13]. Intriguingly, despite this elevated LD_50_, no L925V mutations were detected, suggesting that resistance may arise through alternative mechanisms, such as metabolic detoxification or target-site modifications beyond the L925 locus. These trends predict a continued escalation in LD_50_ values unless resistance management strategies are implemented. If sensitive individuals had been identified in our study, appropriate comments could have been made on the phenotypic resistance of resistant and sensitive individuals according to the L50 level, and the contribution of the change in genotype to the phenotype could have been assessed. Longitudinal monitoring of LD_50_ levels, as demonstrated in New Zealand, provides a phenotypic framework to track resistance evolution over time. Such data, combined with mutational profiling, could clarify the contribution of specific amino acid substitutions to resistance severity.

Synthetic pyrethroids (e.g., flumethrin, tau-fluvalinate) have been extensively used to control *Varroa destructor* mites, a major contributor to honeybee colony losses. Resistance in this mite was first reported in the Lombardy region of Italy during the 1990s [21]. Studies have identified three distinct mutations in the gene encoding the voltage-gated sodium channel (VGSC) protein of varroa mites at leucine position 925 (L925) in flumethrin-resistant populations [17], specifically L925V, L925I, and L925M substitutions [17]. To address the need for the rapid detection of these mutations across populations, a PCR-RFLP assay was developed, enabling widespread screening for pyrethroid resistance. Given the limited arsenal of effective acaricides, this molecular approach provides a critical tool for determining pyrethroid resistance status in global varroa management programs. Long-term control strategies now necessitate routine resistance monitoring to inform adaptive mitigation efforts.

These findings confirm that mutations arose at the drug’s target site across the entire population, corroborated by sequence analysis. Of the 85 mites analyzed from 17 locations, RFLP profiling of 5 pooled samples revealed heterozygous (SR) genotypes (bands at 701-, 438-, and 263-bp) alongside putative homozygous resistant (RR) individuals. The absence of homozygous susceptible (SS) genotypes via RFLP necessitated Sanger sequencing for validation. Sequencing unambiguously identified alleles, and heterozygous allele frequencies were quantified through visual inspection and ImageJ analysis, as described by Stara et al. (2019).

Surprisingly, most mites exhibited a resistance profile. Specifically, 95% carried resistant alleles (R; L925V/I/M), while only 5% retained the susceptible allele (S; Leu). Among the resistant alleles, 91% encoded isoleucine (Ile), 3% valine (Val), and 6% methionine (Met). Notably, no homozygous susceptible (SS) mites were detected, and Ile predominated among the resistance-associated mutations. Global studies have identified L925I, L925V, and L925M substitutions in the flumethrin-targeted VGSC gene region [1,4,5,6,8,12,17,22,23,24,25]. These mutations, first reported in Lombardy, Italy, are hypothesized to have disseminated globally [21,26]. In Lombardy, early studies documented 11% homozygous resistant (RR) and 2% heterozygous (SR) individuals for L925I, with susceptible (SS) mites still present [21,26]. In contrast, our study detected near-complete resistance fixation, with 95% RR and 5% SR genotypes and no SS individuals—reflecting a dramatic rise in resistance prevalence.

Similar to our findings, Benito et al. (2018) reported widespread homozygosity and all three mutations in Spain [1]. In the U.S., Millán-Leiva et al. (2021) also linked flumethrin resistance to L925 substitutions [8]. Regional variations exist: Alissandrakis et al. (2017), in Greece, reported higher L925I prevalence (54% RR) [27], while Vlogiannitis et al. (2021), in Belgium, observed L925V-driven resistance (LC_50_ increased 12.64-fold) [6]. Globally, L925V dominates in Europe [23], whereas L925I and L925M are prominent in Iran [24], Canada [28], and Argentina/Uruguay [19].

In Türkiye, earlier studies detected L925V/I (75% resistance) in Muğla but retained susceptible individuals [16], consistent with national findings of 69% resistance [14,15]. However, a 2021 Turkish study reported 91% resistance [15], aligning with our results. Notably, our study newly identified L925M (6%) in Muğla, a mutation absent in prior regional reports [14,15,16]. Discrepancies in mutation frequencies exist; while L925I dominated here (91%), a 2024 Turkish study found L925I in 21% and L925M in 2.8% of mites [14]. Critically, no Turkish studies—including ours—correlated genotypic resistance with phenotypic LD_50_ data from susceptible populations, limiting mechanistic insights.

The absence of susceptible (SS) varroa samples precluded a direct comparison of LD_50_ values between resistant and susceptible populations. Consequently, the phenotypic implications of molecularly confirmed resistance (e.g., clinical efficacy thresholds) remain unclear. Notably, substitutions at the VGSC L925 locus (Leu → Ile/Met/Val) showed no correlation with LD_50_ variability, suggesting these mutations confer comparable resistance levels regardless of the substituted amino acid. Furthermore, point mutations identified in non-target regions (e.g., positions 423, 147, 149, 150) exhibited no measurable impact on phenotypic resistance, as reflected in the unchanged LD_50_ values.

## 5. Conclusions

The findings of this study demonstrate widespread genotypic resistance to flumethrin in *Varroa destructor* populations, corroborated by elevated LD_50_ values indicative of phenotypic resistance. The combined evidence suggests that flumethrin applications are likely insufficient for effective varroa control, potentially exacerbating colony losses due to sublethal field concentrations. In response, beekeepers may resort to off-label, high-dose flumethrin use, heightening risks of colony collapse and pesticide contamination in hive products. Furthermore, the near-complete fixation of resistant alleles (95% RR, 5% SR) suggests that remaining susceptibility will rapidly erode, as residual susceptible (S) alleles are under intense selection pressure.

To mitigate resistance-driven colony declines, we recommend phasing out flumethrin in regions with confirmed resistance and adopting integrated pest management (IPM) strategies. In areas where efficacy persists, rotational use with non-pyrethroid acaricides is critical to delay resistance evolution. Despite widespread resistance, dose-dependent mortality was observed, implying potential for optimized dosing regimens. However, urgent research is needed to (1) refine flumethrin application protocols, (2) develop novel control agents, and (3) characterize the functional role of non-target mutations (e.g., positions 423, 147, 149, 150) in resistance mechanisms.

## Figures and Tables

**Figure 1 insects-16-00548-f001:**
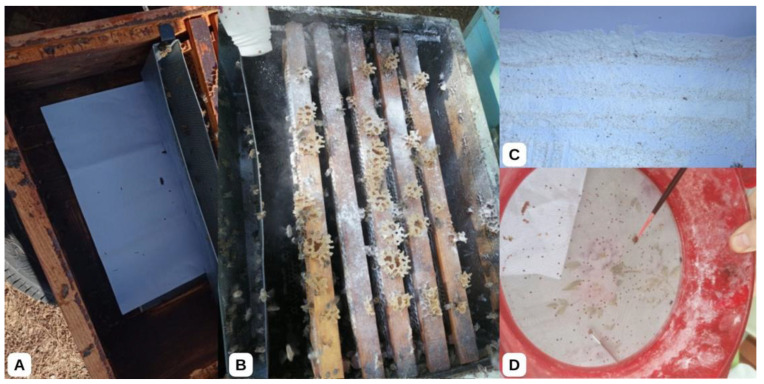
Collection of live varroa samples from the beehives. (**A**) Laying A3 paper under the frames; (**B**) applying powdered sugar between the frames on the honeybees; (**C**) collecting varroa falling on the paper; (**D**) sieving the powdered sugar with a sieve to clean the varroa.

**Figure 2 insects-16-00548-f002:**
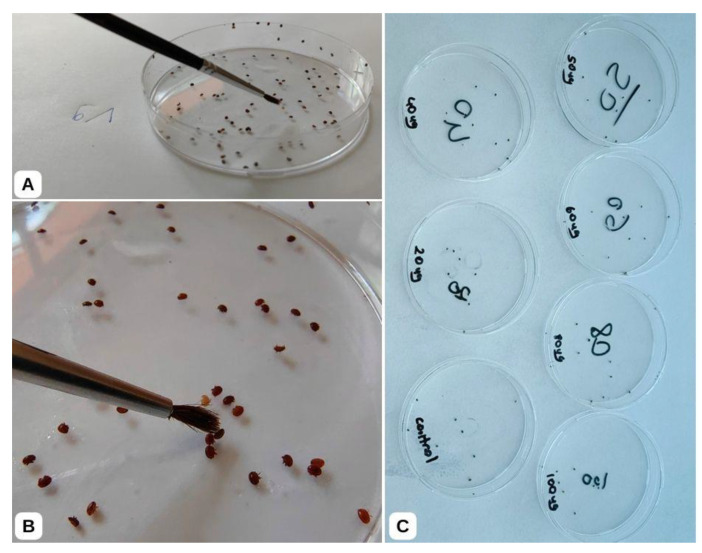
The LD_50_ analyses of varroa mite samples were performed in petri dishes. (**A**) The varroa mites cleaned before were removed with a paintbrush into a clean petri dish; (**B**) varroa mites were kept for 2 h and then checked for viability with a paintbrush touch; (**C**) active varroa mites were transferred to petri dishes loaded beforehand with different concentrations of flumethrin for LD_50_ tests.

**Figure 3 insects-16-00548-f003:**
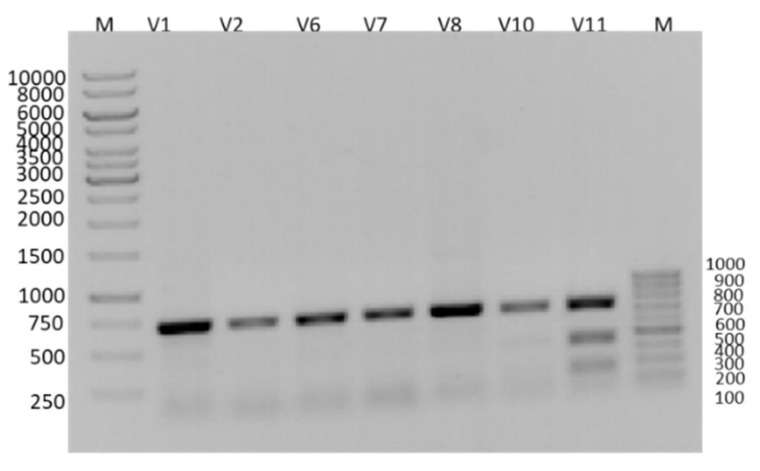
Agarose gel image of varroa PCR products cut with SacI restriction enzyme for RFLP analysis.

**Figure 4 insects-16-00548-f004:**
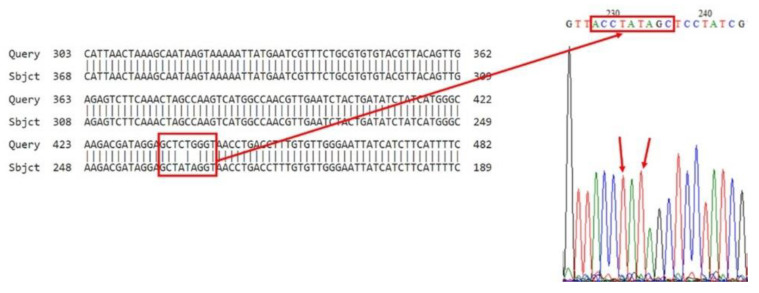
Bioinformatic analysis of the sequence result of sample V6.

**Figure 5 insects-16-00548-f005:**
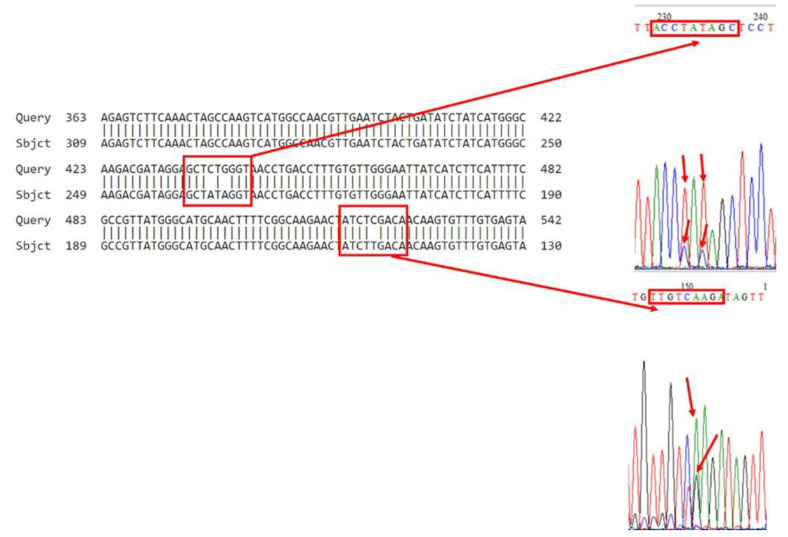
Bioinformatic analysis of the sequence result of sample V8.

**Table 1 insects-16-00548-t001:** LD_50_ results of varroa samples collected from apiaries and mortality by dose.

Apiaries	Location	LD_50_ (µg)	DOSES µg/petri (Diameter = 8.5 cm)						
			10	20	40	50	60	80	100
			Average Mortality ^a^						
V1	Menteşe	42	0	2	4	7	8	10	10
V2	Ula	39.2	2.1	2	5.3	6	8.5	10	10
V3	Milas	61.5	0	0	2	3	5	5	10
V4	Milas	46	0	1	4	6	7	10	10
V5	Milas	51.5	0	1	3	5	6	8.2	10
V6	Milas	51.4	0	1	1	6	6.4	8	10
V7	Bodrum	44.6	0	2	4	6	8	8.2	10
V8	Milas	36.5	0	3	6	8	8	10	10
V9	Menteşe	48	0	1	3	4	8	10	10
V10	Menteşe	47.5	0	1	2	6	9	10	10
V11	Köyceğiz	53.6	0	0	1.8	2.5	8	10	10
V12	Milas	38.4	0.7	3.1	4.7	6.5	8.3	10	10
V13	Milas	45	0	1	3	7	8	10	10
V14	Milas	54.5	0	0	2.5	4.2	5	10	10
V15	Milas	61.8	0	0	1.3	2	5	8.6	10
V16	Milas	44.4	1	1.7	2.5	5	8	10	10
V17	Köyceğiz	54.2	1	0.9	1.3	5	7	10	10
V18	Köyceğiz	57.5	0	0	1.5	2.7	5.5	10	10
V19	Milas	45.5	0.9	0.7	4.7	6.3	6	9.3	10
V20	Ula	60.2	0	0	3	3	4	8	10
V21	Ula	54.3	0	1.4	2.9	3.8	7.3	6.9	10
V22	Dalaman	31	1	2	7	10	10	10	10
V23	Dalaman	50	0	1.7	3.3	4	6	10	10
V24	Dalaman	47	0	2	4	5	6	10	10
V25	Dalaman	46.5	0	2	3	4	8	10	10
V26	Dalaman	61	0	1	2	2	3	8	10
V27	Dalaman	53.5	0	0	3	5	6	8	10
		49.13	0.24	1.17	3.18	4.99	6.85	9.19	10

^a^ Average mortality: Provides the average mortality resulting from the standardisation of the mortality of 8–13 varroa used at each dose to 10 varroa.

**Table 2 insects-16-00548-t002:** Evaluation of VGSC profiles according to RFLP results and allele probabilities of varroa samples.

VGSC Alleles	Varroa Samples	Genotype Probabilities
L/L homozygous susceptible alleles	-	SS (Leu/Leu)
V/V homozygous resistant alleles	V1, V2, V3, V4, V5 V6, V7, V8, V9, V12, V13, V14, V15, V16, V19, V21, V22	RR (Ile/Ile, Ile/Met, Ile/Val, Val/Val, Val/Met, Met/Met)
L/V heterozygous alleles (some of 5 samples)	V10x, V11x, V17x, V18x, V20x	SR (Leu/X resistant allele)

**Table 3 insects-16-00548-t003:** Determination of alleles of varroa (V) samples according to RFLP and sequence results.

No ^a, b^	VGSC Alleles	No ^a,b^	VGSC Alleles	No ^a,b^	VGSC Alleles	No ^a,b^	VGSC Alleles
V1a V1b V1c V1d V1e	5 samples Homozygote RR (Ile/Ile) (CTG→ATA) (5 RR)	V7a V7b V7c V7d V7e	5 samples Homozygote RR (Ile/Ile) (CTG→ATA) (5 RR)	V13a V13b V13c V13d V13e	5 samples Homozygote RR allele (5 RR)	V19a V19b V19c V19d V19e	5 samples Homozygote RR allele (5 RR)
V2a V2b V2c V2d V2e	5 samples Homozygote RR (Ile/Ile) (CTG→ATA) (5 RR)	V8a V8b V8c V8d V8e	Of the 10 alleles, 70% were RR (Ile/Ile), and 30% were GTG (Val) alleles (RR). (5 RR)	V14a V14b V14c V14d V14e	5 samples Homozygote RR allele (5 RR)	V20a V20b V20c V20d V20e	Of the 10 alleles, 70% were RR (Ile/Ile), 20% were ATG RR (Met) allele (RR), and 10% were CTG (Leu) S allele. (4 RR and 1SR)
V3a V3b V3c V3d V3e	5 samples Homozygote RR Allel (5 RR)	V9a V9b V9c V9d V9e	Of the 10 alleles, 70% were RR (Ile/Ile), and 30% were ATG (Met) alleles (RR). (5 RR)	V15a V15b V15c V15d V15e	5 samples Homozygote RR allele (5 RR)	V21a V21b V21c V21d V21e	5 samples Homozygote RR allele (5 RR)
V4a V4b V4c V4d V4e	5 samples Homozygote RR (Ile/Ile) (CTG→ATA) (5 RR)	V10a V10b V10c V10d V10e	Of the 10 alleles, 60% were RR (Ile/Ile), 20% were ATG RR (Met) allele (RR), and 20% were CTG (Leu) S allele. 3 RR, 2SR or 4RR, 1SS	V16a V16b V16c V16d V16e	5 samples Homozygote RR allele (5 RR)	V22a V22b V22c V22d V22e	5 samples Homozygote RR (Ile/Ile) (CTG→ATA) (5 RR)
V5a V5b V5c V5d V5e	5 samples Homozygote RR (Ile/Ile) (CTG→ATA) (5 RR)	V11a V11b V11c V11d V11e	Of the 10 alleles, 60% were RR and 40% were CTG (Leu) S alleles 2SS+3RR or 4SR+1RR or 1SS+2SR+2RR	V17a V17b V17c V17d V17e	Of the 10 alleles, 80% were R, and 20% were CTG (Leu) S alleles. 3 RR, 2SR or 4RR, 1SS	The total number of mites being pooled was 22 × 5 = 110. There were 220 alleles for 110 diploid mites. When allele frequencies were evaluated, 95% of the alleles detected were resistant, while 5% of the alleles detected were sensitive. The populations were highly resistant to the drug.	
V6a V6b V6c V6d V6e	5 samples Homozygote RR (Ile/Ile) (CTG→ATA) (5 RR)	V12a V12b V12c V12d V12e	5 samples Homozygote RR allele (5 RR)	V18a V18b V18c V18d V18e	10 allelden %80’i R alleli, %20’si CTG (Leu) S alleli. 3 RR, 2SR veya 4RR, 1SS.		

^a^ Samples V1, V2, V4, V5, V6, V7, V8, V9, V10, V20, and V22 were sequenced. ^b^ V3, V11–19, and V21 samples were evaluated according to the RFLP results.

## Data Availability

The original contributions presented in this study are included in the article/Appendix A. Further inquiries can be directed to the corresponding author(s).

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
