# Peer review of "Assessing Phenotypic and Genotypic Resistance to Flumethrin in Varroa destructor Populations in Muğla, Türkiye"

_insects, 2025, doi:10.3390/insects16060548_

Round 1

Reviewer 1 Report

Comments and Suggestions for Authors

The idea behind this study is highly relevant, as it addresses a recurrent and pressing issue: Varroa resistance to flumethrin. Honeybee colony losses have been consistently reported over the years, and once again, this year appears to be devastating for honeybee populations worldwide. One of the main causes consistently pointed out is the Varroa mite. As such, this study, even if conducted at a regional level, has the potential to offer valuable insights into the current status of resistance to flumethrin and could help, for example, beekeepers develop better strategies to protect their colonies.

However, there are significant methodological flaws that need to be addressed before this work can be considered for publication. In particular, the lack of a proper control group and the absence of appropriate statistical analyses seriously undermine the reliability of the results. These issues must be carefully revised and corrected to ensure the validity and robustness of the study's conclusions (see suggestions below).

Regarding the methodology, this manuscript lacks several critical elements, particularly in terms of the detail required for reproducibility. Throughout the M&M section, important information is either missing or insufficiently described. I would also suggest the authors use consistent concentration units throughout the text (mg or ul). It is further mentioned that flumethrin was prepared using Bayticol, but no further details are provided on how this was done, making it difficult to replicate the procedure. One of the major issues is the lack of a known sensitive/susceptible strain of Varroa mites that have not been exposed to flumethrin or similar acaricides, to serve as a baseline and control group for comparing LDâ‚…â‚€ values. If this is not currently feasible, I would suggest comparing their results with published LDâ‚…â‚€ values for susceptible Varroa populations available in the literature. This comparison is crucial to establish a threshold for distinguishing between resistance and susceptibility based on LDâ‚…â‚€ values.

In this same section of M&M, no statistical analyses are mentioned. In the first part of the Results section (lines 182–183), the authors explain how the LDâ‚…â‚€ was calculated using the Karber-Behrens method. However, this description should be included in the M&M section instead.  Moreover, the statistical method employed is not the most appropriate for this type of analysis, and no confidence intervals for the LDâ‚…â‚€ values are provided. Since the aim is to estimate LDâ‚…â‚€ and assess how mortality depends on the dose, logistic regression models would be more suitable. I suggest using a Binomial Generalized Linear Model (GLM), with binary response data (dead/alive) and dose as a continuous predictor. This approach allows for the estimation of the dose at which the probability of mortality is p=0.5 (the LDâ‚…â‚€), and it can be implemented using statistical software such as R. Furthermore, it is not clearly stated in the text whether replicates were used in the LDâ‚…â‚€ assays. In the M&M section, the authors mention, "After assessing viability… ten of them were placed in Petri dishes…” suggesting that a single group of 10 mites per dose may have been tested without replication. However, in Table 1, each dose appears to be linked to an *average* mortality of 10 varroa per dose, which could imply that replicates were indeed used. The authors should specify how many replicate Petri dishes were tested per dose, and whether the mortality data in Table 1 reflect single or averaged replicates.

One of the goals of the authors was also to assess phenotypic and genotypic resistance, however, this cannot be fully determined due to the lack of sensitive control, as it is not possible to infer if mortality rates are due to the resistance or susceptibility of of the individuals. Finally, one of the major goals of this study is to connect the VGSC mutations to resistance, i.e., a correlation between genotype and phenotype, could not be statistically validated, as all the individuals used for the molecular analyses were found to be resistant, issued that was also raised by the authors in the discussion.

Based on my comments, I suggest the authors revise the statistical analysis approach and address the above methodological issues. These changes would greatly improve the quality of the manuscript. Once more, I emphasize that this study's findings could be highly relevant and provide important insights into the current status of Varroa's resistance to flumethrin.

Author Response

Referee 1

The idea behind this study is highly relevant, as it addresses a recurrent and pressing issue: Varroa resistance to flumethrin. Honeybee colony losses have been consistently reported over the years, and once again, this year appears to be devastating for honeybee populations worldwide. One of the main causes consistently pointed out is the Varroa mite. As such, this study, even if conducted at a regional level, has the potential to offer valuable insights into the current status of resistance to flumethrin and could help, for example, beekeepers develop better strategies to protect their colonies.

However, there are significant methodological flaws that need to be addressed before this work can be considered for publication. In particular, the lack of a proper control group and the absence of appropriate statistical analyses seriously undermine the reliability of the results. These issues must be carefully revised and corrected to ensure the validity and robustness of the study's conclusions (see suggestions below).

Regarding the methodology, this manuscript lacks several critical elements, particularly in terms of the detail required for reproducibility. Throughout the M&M section, important information is either missing or insufficiently described.

 I would also suggest the authors use consistent concentration units throughout the text (mg or ul).

Response: Since the prepared concentrations were low, µg was used. Therefore, we changed the mg values in the text to µg.

 It is further mentioned that flumethrin was prepared using Bayticol, but no further details are provided on how this was done, making it difficult to replicate the procedure.

Response: The procedure is described between L105-L115 in M&M section.

One of the major issues is the lack of a known sensitive/susceptible strain of Varroa mites that have not been exposed to flumethrin or similar acaricides, to serve as a baseline and control group for comparing LDâ‚…â‚€ values. If this is not currently feasible, I would suggest comparing their results with published LDâ‚…â‚€ values for susceptible Varroa populations available in the literature. This comparison is crucial to establish a threshold for distinguishing between resistance and susceptibility based on LDâ‚…â‚€ values.

Response: These susceptible individuals could not be identified in this study.  It is also not possible to provide standardised sensitive individuals.

Response: All previous LD 50 studies of flumethrin were discussed and summarised. The discussion is given between L285-L305.

In this same section of M&M, no statistical analyses are mentioned. In the first part of the Results section (lines 182–183), the authors explain how the LDâ‚…â‚€ was calculated using the Karber-Behrens method. However, this description should be included in the M&M section instead.

Response: This section is taken to ‘Acute LDâ‚…â‚€ analysis and determination of phenotypic resistance’ in M&M.

  Moreover, the statistical method employed is not the most appropriate for this type of analysis, and no confidence intervals for the LDâ‚…â‚€ values are provided. Since the aim is to estimate LDâ‚…â‚€ and assess how mortality depends on the dose, logistic regression models would be more suitable. I suggest using a Binomial Generalized Linear Model (GLM), with binary response data (dead/alive) and dose as a continuous predictor. This approach allows for the estimation of the dose at which the probability of mortality is p=0.5 (the LDâ‚…â‚€), and it can be implemented using statistical software such as R. Furthermore, it is not clearly stated in the text whether replicates were used in the LDâ‚…â‚€ assays. In the M&M section, the authors mention, "After assessing viability… ten of them were placed in Petri dishes…” suggesting that a single group of 10 mites per dose may have been tested without replication. However, in Table 1, each dose appears to be linked to an *average* mortality of 10 varroa per dose, which could imply that replicates were indeed used. The authors should specify how many replicate Petri dishes were tested per dose, and whether the mortality data in Table 1 reflect single or averaged replicates.

Response: There are different valid methods for acute LD50 tests. The Karber-Behrens method is one of those used in scientific studies. I have been using the Karber-Behrens method before, so I have knowledge of the test. I do not have experience with the test mentioned by the referee, therefore I could not use it. In this study, at least 70 varroa samples are required for each apiary. Usually only half of the collected varroa passed the viability tests, so an average of 100-150 varroa samples were collected from each apiary. Due to the lack of viable varroa samples, each dose could not be repeated.  All samples collected were resistant. Therefore, the significance test of the difference between resistant and resistant samples could not be performed. If sensitive individuals had been identified, the difference between them would have been analysed by Chi-squared test.

One of the goals of the authors was also to assess phenotypic and genotypic resistance, however, this cannot be fully determined due to the lack of sensitive control, as it is not possible to infer if mortality rates are due to the resistance or susceptibility of of the individuals. Finally, one of the major goals of this study is to connect the VGSC mutations to resistance, i.e., a correlation between genotype and phenotype, could not be statistically validated, as all the individuals used for the molecular analyses were found to be resistant, issued that was also raised by the authors in the discussion.

Based on my comments, I suggest the authors revise the statistical analysis approach and address the above methodological issues. These changes would greatly improve the quality of the manuscript. Once more, I emphasize that this study's findings could be highly relevant and provide important insights into the current status of Varroa's resistance to flumethrin.

Response:  I would like to thank the referee for his valuable contributions. The referee actually wants to direct us to the point we are aiming for, but our hands are tied by our inability to identify sensitive individuals in the study. In fact, the aim was to compare the LD50 values of resistant individuals and sensitive individuals and to examine the difference statistically. As a result, it was aimed to draw a path for beekeepers. However, the dramatic results we obtained showed us that we are on the verge of a different danger. I hope that we have been able to give an explanatory answer to the criticisms of the referee.

Reviewer 2 Report

Comments and Suggestions for Authors

General comments on the manuscript

The manuscript contains valuable information on the increased resistance to pesticides exhibited by the varroa mite.

The laboratory tests were carefully conducted but the results could be interpreted better, especially given the current issues with overwintering mortality experienced in the USA and the concerns about pesticide resistance by the mite.

The discussion needs to be organized better, I would suggest that the authors begin by summarizing their results (present and explain). Explain by comparing to previously published works on the same topic. I think you already have numerous recent references you can use.

Work through your discussion in order – topic by topic.

In addition:

  • There are sections that appear to need some English language revisions
  • Consistency of nomenclature: terms such as Varroa are sometimes capitalized (Line 280) and sometimes not (288)

Simple Summary

Line 17 – Flumethrin is the most widely used pesticide among the treatments.

Reword …Flumethrin is among one the most frequently used varroacides.

Please check because it is my understanding that due to high level of resistance to pyrethroids, such as Apistan, the US beekeepers now prefer Amitraz although there is evidence that resistance is also developing against this chemical.

This study compares the effectiveness of 3 miticides including amitraz, tau-fluvalinate, and flumethrin. Both pyrethroids, flumethrin and tau-fluvalinate, were considered mildly effective (<80%).

https://www.cambridge.org/core/services/aop-cambridge-core/content/view/0BCF06498A566F637659CE8CB02D03EE/S0008347X22000049a.pdf/surveillance-of-synthetic-acaricide-efficacy-against-varroa-destructor-in-ontario-canada.pdf

https://honeybeehealthcoalition.org/wp-content/uploads/2021/06/Commercial_Beekeeping_062121.pdf

Abstract

Line 27 – I believe Amitraz is not considered a pyrethroid

Materials and Methods

Line 94 …fallen varroes Should be corrected to …fallen varroas

Lines 94 to 97 Possible reword to

The collection paper on the hive floor was carefully removed 5 minutes after the application of the powdered sugar. A stainer was used to separate powdered sugar residues from the fallen mites (Figure 1). Excess sugar was then dusted off the mites using a paintbrush. The mites were then washed (I think this requires a bit more explanation of how much water was used or how they were rinsed) and finally dried thoroughly with a towel napkin. (paper towel?).

Lines 97-100

To ensure that the collected mites had survived the collection and cleaning procedures they were held in petri dishes (*you may want to indicate how many mites per petri dish) with a small paper towel inside. Only the varroa mites that remained alive after 3 hours post-collection procedures were used in the LD50 test (Figure 2). After completing the LD50 tests the varroa mites were stored at -20cC for molecular analysis.

In the legend of Figure 2 you explain testing viability. This procedure may be best integrated into the materials and methods. You mention “check for viability with a brush”- does this mean that the mites were touched with the brush to test for reactivity? Is this after cleaning before the test? Or after the test to determine their survival after exposure to the pesticide?

Line 113 – Hegzan? Do you mean Hexane?

 Lines 130-132 – please define what is a sample … 5 samples from each location? Is a sample a mite? A group of mites?

Were mites from different apiaries were pooled?

Line 135 – Five varroa individuals were collected to represent each location

Does this mean that a sample is 5 mites per apiary?

 Line 136- replace “hard skin” with “hard exoskeleton” or “tough exoskeleton”

Line 156 – 158 : Please reword: These primer sequences were successfully tested by Millan-Leiva et al. (2018) …

Is the rest of the paragraph indicating that you used the same protocol? If so please reword.

Lines 165-171: Please reword. Maybe some of the descriptions of protocols could be simplified by indicating which previously used protocols are being followed or modified.

Results

Lines 182-185:  should be included as Materials and Methods.

Lines 185 to 201: The paragraph needs to be organized better. There are statements in the paragraph that belong to the discussion, not results.

 For example:

One of the aims of the study 189 was to compare varroa samples that were resistant or sensitive to the molecular results in 190 order to interpret these results as meaningful. However, the molecular results showed 191 that most of the molecular results were homozygous resistance, and very few were heterozygous. As a result, mutations occurred at the target site of the drug in the entire popu-193 lation. Sequence analysis also confirmed these results.

Line 216 – there is an extra period.

Lines 216-217 – Another sentence that should be used in the discussion rather than results. “It was surprising to find that most mites displayed a resistant pattern.”

Discussion

Lines 291-292: Please reword or clarify the following statement: …dose required for flumethrin to have an effect on varroa on the bee was …

Line 292-295: Reword.

Lines 303-305 Here you present a very interesting fact that is somewhat buried in the discussion.

Comments on the Quality of English Language

There are some issues with the quality of English. I would suggest that you get help from a native speaker to make the paragraphs flow better.

Author Response

I would like to thank the referee for his/her valuable contributions and guidance, which were really important for the improvement of the manuscript.

Referee 2

General comments on the manuscript

The manuscript contains valuable information on the increased resistance to pesticides exhibited by the varroa mite.

The laboratory tests were carefully conducted but the results could be interpreted better, especially given the current issues with overwintering mortality experienced in the USA and the concerns about pesticide resistance by the mite.

The discussion needs to be organized better, I would suggest that the authors begin by summarizing their results (present and explain). Explain by comparing to previously published works on the same topic. I think you already have numerous recent references you can use.

Work through your discussion in order – topic by topic.

In addition:

  • There are sections that appear to need some English language revisions

Response: The manuscript has been generally reorganised.

  • Consistency of nomenclature: terms such as Varroa are sometimes capitalized (Line 280) and sometimes not (288)

Response: Re-arrangement was made.

Simple Summary

Line 17 – Flumethrin is the most widely used pesticide among the treatments.

Reword …Flumethrin is among one the most frequently used varroacides.

Response: Re-arrangement was made.

Please check because it is my understanding that due to high level of resistance to pyrethroids, such as Apistan, the US beekeepers now prefer Amitraz although there is evidence that resistance is also developing against this chemical.

I mentioned the articles associated with the topic in the manuscript discussion. The mechanism of action of amitraz is different and therefore there has been no additional evaluation.

This study compares the effectiveness of 3 miticides including amitraz, tau-fluvalinate, and flumethrin. Both pyrethroids, flumethrin and tau-fluvalinate, were considered mildly effective (<80%).

https://www.cambridge.org/core/services/aop-cambridge-core/content/view/0BCF06498A566F637659CE8CB02D03EE/S0008347X22000049a.pdf/surveillance-of-synthetic-acaricide-efficacy-against-varroa-destructor-in-ontario-canada.pdf

https://honeybeehealthcoalition.org/wp-content/uploads/2021/06/Commercial_Beekeeping_062121.pdf

Response: The references mentioned by the referee are cited in the introduction.

Abstract

Line 27 – I believe Amitraz is not considered a pyrethroid

Response: Incorrect information rectified.

Materials and Methods

Line 94 …fallen varroes Should be corrected to …fallen varroas

Response: We have corrected it to fallen varroa mites according to the other referee's suggestion.

Lines 94 to 97 Possible reword to

The collection paper on the hive floor was carefully removed 5 minutes after the application of the powdered sugar. A stainer was used to separate powdered sugar residues from the fallen mites (Figure 1). Excess sugar was then dusted off the mites using a paintbrush. The mites were then washed (I think this requires a bit more explanation of how much water was used or how they were rinsed) and finally dried thoroughly with a towel napkin. (paper towel?).

Response: Arrangement was made in accordance with the referee's recommendation.

Lines 97-100

To ensure that the collected mites had survived the collection and cleaning procedures they were held in petri dishes (*you may want to indicate how many mites per petri dish) with a small paper towel inside. Only the varroa mites that remained alive after 3 hours post-collection procedures were used in the LD50 test (Figure 2). After completing the LD50 tests the varroa mites were stored at -20cC for molecular analysis.

Response: Arrangement was made in accordance with the referee's recommendation.

In the legend of Figure 2 you explain testing viability. This procedure may be best integrated into the materials and methods. You mention “check for viability with a brush”- does this mean that the mites were touched with the brush to test for reactivity? Is this after cleaning before the test? Or after the test to determine their survival after exposure to the pesticide?

Response: Edited and added the required explanation.

Line 113 – Hegzan? Do you mean Hexane?

Response: Incorrect information rectified.

 Lines 130-132 – please define what is a sample … 5 samples from each location? Is a sample a mite? A group of mites?

Response: Edited and added the required explanation.

Were mites from different apiaries were pooled?

Response: Varroa samples collected from hives in each location, not different apiaries, were pooled.

Line 135 – Five varroa individuals were collected to represent each location

Does this mean that a sample is 5 mites per apiary?

Response: Yes.

 Line 136- replace “hard skin” with “hard exoskeleton” or “tough exoskeleton”

Response: Replaced with tough exoskeleton.

Line 156 – 158 : Please reword: These primer sequences were successfully tested by Millan-Leiva et al. (2018) …

Is the rest of the paragraph indicating that you used the same protocol? If so please reword.

Response: Arrangements have been made

Lines 165-171: Please reword. Maybe some of the descriptions of protocols could be simplified by indicating which previously used protocols are being followed or modified.

Response: Arrangements have been made.

Results

Lines 182-185:  should be included as Materials and Methods.

Response: Moved to the material method ‘Acute LDâ‚…â‚€ analysis and determination of phenotypic resistance’.

Lines 185 to 201: The paragraph needs to be organized better. There are statements in the paragraph that belong to the discussion, not results.

 For example:

One of the aims of the study 189 was to compare varroa samples that were resistant or sensitive to the molecular results in 190 order to interpret these results as meaningful. However, the molecular results showed 191 that most of the molecular results were homozygous resistance, and very few were heterozygous. As a result, mutations occurred at the target site of the drug in the entire popu-193 lation. Sequence analysis also confirmed these results.

Response: The sections belonging to the discussion have been moved to the end of the discussion.

Line 216 – there is an extra period.

Response: Could not be understood.

Lines 216-217 – Another sentence that should be used in the discussion rather than results. “It was surprising to find that most mites displayed a resistant pattern.”

Response: It was removed.

Discussion

Lines 291-292: Please reword or clarify the following statement: …dose required for flumethrin to have an effect on varroa on the bee was …

Response: Arrangements have been made.

Line 292-295: Reword.

Response: Arrangements have been made.

Reviewer 3 Report

Comments and Suggestions for Authors

While this seems to be a useful study, it is not at all well presented and requires clarification and explanation in various places. In places it is confusing.

In the title and elsewhere “Turkiye” should be changed to “Turkey”, for this international English language journal.

The abstract is long and the length should be checked.

In Methods, line 86 onwards, it should be mentioned when the study took place- when was the sample collection done?

L107: 3 apiaries are mentioned here but 27 were involved in the study? This needs to be clarified.

Also, it seems that 27 sampling locations (apiaries) were used but the paper often refers to 22. This is confusing and must be clarified or corrected.

L113: I think “Hegzan” should be “hexane”.

Figure 2 legend: some rewording is needed. Also “petri” and “Petri” are both used here.

L130, 134, section 2.6, L240, L328 and elsewhere: here you refer to apiaries V1-V22. What about the others? There were 27.

L165-166: these are not clear and are not in sentences and must be corrected.

In the methods generally, each section should be reviewed. There is some repetition and it seems that the material is not always given in a logical order.

In Results, the authors MUST remove the column in Table 1 which names the beekeeper- including this is completely inappropriate.  Also it is unclear what the part marked in brackets after line 202 means.

L208: this heading is not properly worded for a heading.

L273 needs clarification.

In the Discussion, L291-292, some rewording is suggested to give the correct meaning.

L361-363 and L364-365 are unclear and need reworded.

L382 onwards, in various places studies are referred to which need a reference cited close to where they are mentioned, to be clear which study is being discussed. The words “this study” are used repeatedly, without it being clear which study is concerned. See the annotated text.

L407: it is unclear what is meant by “flumetry”. Can this be reworded?

The references need some attention. See the annotated text.

The paper generally needs considerable tidying up. As well as the above points, various changes to wording are needed and I have marked these on the annotated document for the attention of the authors.  The style is inconsistent in various ways, e.g. use of LD50 and LD subscript 50, Varroa and varroa, Flumethrin and flumethrin. The style of in-text referencing and the listed references also varies.  Some sentences are not needed and have been marked.

Comments on the Quality of English Language

The wording needs improved in various places. Please see the annotated document.

Author Response

I would like to thank the referee for his/her valuable contribution. The requested edits have been made in the pdf text. In addition, the following change requests have been edited.

Referee  3

While this seems to be a useful study, it is not at all well presented and requires clarification and explanation in various places. In places it is confusing.

In the title and elsewhere “Turkiye” should be changed to “Turkey”, for this international English language journal.

Response: The United Nations (UN) changed the name of the country, which is used as ‘Turkey’ in foreign languages, to ‘Türkiye’.

https://www.un.org/en/about-us/member-states/turkiye

The abstract is long and the length should be checked.

Response: The summary has been reorganised and shortened as far as possible.

In Methods, line 86 onwards, it should be mentioned when the study took place- when was the sample collection done?

Response: Dates added.

L107: 3 apiaries are mentioned here but 27 were involved in the study? This needs to be clarified.

Response: For the preliminary study mentioned here, 3 apiaries were used and then field work was carried out. therefore ‘preliminary testing’ was written in the sentence.

Also, it seems that 27 sampling locations (apiaries) were used but the paper often refers to 22. This is confusing and must be clarified or corrected.

Response: 27 apiaries were studied and LD50 tests were performed in 27 of them. However, the tests obtained from 5 apiaries were not included in the molecular study due to insufficient DNA isolation. Due to the change of season, we have not had a chance to resample.

  ‘DNA Isolation from Varroa samples’ a description has been added to the section.

L113: I think “Hegzan” should be “hexane”.

Response: Typo corrected.

Figure 2 legend: some rewording is needed. Also “petri” and “Petri” are both used here.

Response: Figure 2 title and subtitle have been rearranged.

L130, 134, section 2.6, L240, L328 and elsewhere: here you refer to apiaries V1-V22. What about the others? There were 27.

Response: Information provided above.

L165-166: these are not clear and are not in sentences and must be corrected.

Response: Re-arrangement was made.

In the methods generally, each section should be reviewed. There is some repetition and it seems that the material is not always given in a logical order.

Response: Re-arrangements were made.

In Results, the authors MUST remove the column in Table 1 which names the beekeeper- including this is completely inappropriate.  

Response: Beekeeper names were removed.

Also it is unclear what the part marked in brackets after line 202 means.

Response: The symbol changed to diameter.

L208: this heading is not properly worded for a heading.

Response: Re-arrangements were made.

Sequence analysis of the V1 (V1a, V1b, V1c, V1d, V1e), V2 (V2a, V2b, V2c, V2d, V2e), 238

V4 (V4a, V4b, V4c, V4d, V4e), V5 (V5a, V5b, V5c, V5d, V5e), V6 (V6a, V6b, V6c, V6d, V6e), 239

V7 (V7a, V7b, V7c, V7d, V7e), V22(V22a, V22b, V22c, V22d, and V22e) samples confirmed.

Why choose these ones and not the others?

Response: We selected ‘representative’ samples from samples with homozygous and heterozygous patterns.

L273 needs clarification.

 Response: Re-arrangements were made.

In the Discussion, L291-292, some rewording is suggested to give the correct meaning.

Response: Re-arrangements were made.

L361-363 and L364-365 are unclear and need reworded.

Response: Re-arrangements were made.

L382 onwards, in various places studies are referred to which need a reference cited close to where they are mentioned, to be clear which study is being discussed. The words “this study” are used repeatedly, without it being clear which study is concerned. See the annotated text.

Response: Edited and added the cited source.

L407: it is unclear what is meant by “flumetry”. Can this be reworded?

Response: Re-arrangements were made.

The references need some attention. See the annotated text.

The paper generally needs considerable tidying up. As well as the above points, various changes to wording are needed and I have marked these on the annotated document for the attention of the authors.  The style is inconsistent in various ways, e.g. use of LD50 and LD subscript 50, Varroa and varroa, Flumethrin and flumethrin. The style of in-text referencing and the listed references also varies.  Some sentences are not needed and have been marked.

Response: Edited as follows

 LD50 changed to LDâ‚…â‚€

Flumethrin changed to flumethrin

Varroa changed to varroa

Reviewer 4 Report

Comments and Suggestions for Authors

The idea is not new (the same problem with the same methodology was investigated in references Nos. 15, 16, 17, 19, 20, 21, 22, 23, 24, 25, 26). On the other hand, this investigation is properly done, so I have nothing against its being published. I like the photos as proof of their dedicated work. My comments and suggestions for improvement are given below.

SIMPLE SUMMARY

Line 17: Have you found recent references for this statement: „Flumethrin is the most widely used pesticide among the treatments“? If not, I advise you to avoid such strong statements.

Line 18: Mistake. It is not „In recent years“. It has been 30 years since flumethrin has been known to be ineffective. Check out some of those old references below (and there are plenty of more recent ones, too).

  • Milani N. (1995) The resistance of Varroa jacobsoni to pyrethroids: a laboratory assay, Apidologie 26, 415-429.
  • Milani N (1999) The resistance of Varroa jacobsoni to acaricides. Apidologie 30, 229-234.
  • Thompson HM, Brown MA, Ball RF, Bew MH (2002) First report of Varroa destructor resistance to pyrethroids in the UK. Apidologie, 33, 357–366.
  • Loucif-Ayad W, Aribi N, Smagghe G, Soltani N (2010) Comparative effectiveness of some acaricides used to control Varroa destructor (Mesostigmata: Varroidae) in Algeria. African Entomology 18, 259–266.

  1. INTRODUCTION:

Lines 81-82: I think you should avoid the term „aim“ here. Please rephrase the sentence.

  1. MATERIALS AND METHODS

Line 85:  This is grammatically improper: „The varroa ... were collected“. You may write: varroa mites ... were collected.

Line 97: Delete „towel napkin“ and write either „paper towel“ or „paper napkin“.

Line 98: Delete „towel paper“ and write „paper towel“.

Line 113: „Hegzan“ is probably a misspelling of „Hexane“; please check and correct.

Line 122: I suggest to replace „to analyse the molecular resistance“ with „for further analyses of flumethrin resistance mutations“.

Line 129: In the subtitle 2 3. you should add „DNA“ before „ isolation from varroa samples“. Furthermore, write „Varroa“ instead of „varroa“.

Line 140: Check this: „and vortexing in between“, as it is unclear.

Line 141: Check this „the kit protocol was maintained“, as it is unclear.  

Line 149: After the sentence „The following reaction contents were used in the experiment“ the content is missing. Please correct that.

Line 150: This is improperly written: „RAPD primers were found to be a sufficient amount of ..“ Please check and correct.

Lines 151-152: This sentence is redundant: „According to this result, it was evaluated that primers for drug-resistance genes could be ordered, and the desired gene regions could be amplified and analyzed“. Please delete it.

Line 153-154: This part is redundant: „and this gene was amplified using the DNA obtained“. Please delete it.

Line 155: Insert „[19]“ after the „5′-GCTGTTGTTACCGTGGAGCA-3′.

Lines 156-157: Delete this sentence: „These primer sequences were tested by Millan-Leiva et al. (2018), and successful results were obtained [3]“. The sentence is redundant and wrong since you erroneously referred to the ref. [3]; Simply, Dietemann et al. (2013) could not obtain the results in 2013 with the primers designed in 2018.

Line 158: After the sentence „The following reaction ingredients were used in the PCR experiment“ the ingredients are missing. Please correct that.

Lines 165-166: These two sentences should be merged and corrected, including the reference in square brackets at the end: „RFLP analyzed the obtained PCR products. The technique performed by Millan-Leiva et al. (2018) [3]“. I suggest correcting that as follows: „PCR products were analysed by RFLP established by Millan-Leiva et al. (2018) [19]“.

  1. DISCUSSION

Line 313: Please provide recent references for this statement: „Synthetic pyrethroids (flumethrin, tau-fluvalinate, etc.) are extensively used“.

Lines 328-340: These three paragraphs are not adequate for the Discussion as represent a repetition of the methodology. Please delete it.

Lines 341-352: In this paragraph, there are some results, but most of the text is a repetition of the methodology. Please shorten the paragraph and avoid describing the methodology.

Line 406-407: Something is wrong here „flumetry in the field“. Considering the meaning of the following sentences (Lines 407-411), I am rather sure that the sentence in Lines 406-407 should be deleted. 

Comments on the Quality of English Language

Polishing of the English language would be beneficial. I think I pointed out the biggest mistakes in the comments, but I am sure I missed some.

Author Response

The valuable referee's observations and guidance contributed significantly to the improvement of the manuscript. Therefore, thank you very much to the referee. I have edited the referee's suggestions to a considerable extent.

Referee 4

The idea is not new (the same problem with the same methodology was investigated in references Nos. 15, 16, 17, 19, 20, 21, 22, 23, 24, 25, 26). On the other hand, this investigation is properly done, so I have nothing against its being published. I like the photos as proof of their dedicated work. My comments and suggestions for improvement are given below.

SIMPLE SUMMARY

Line 17: Have you found recent references for this statement: „Flumethrin is the most widely used pesticide among the treatments“? If not, I advise you to avoid such strong statements.

Response: Sentence revised on the referee's suggestion.

Line 18: Mistake. It is not „In recent years“. It has been 30 years since flumethrin has been known to be ineffective. Check out some of those old references below (and there are plenty of more recent ones, too).

Response: Sentence corrected due to the referee's justified warning.

  • Milani N. (1995) The resistance of Varroa jacobsoni to pyrethroids: a laboratory assay, Apidologie 26, 415-429.
  • Milani N (1999) The resistance of Varroa jacobsoni to acaricides. Apidologie 30, 229-234.
  • Thompson HM, Brown MA, Ball RF, Bew MH (2002) First report of Varroa destructor resistance to pyrethroids in the UK. Apidologie, 33, 357–366.
  • Loucif-Ayad W, Aribi N, Smagghe G, Soltani N (2010) Comparative effectiveness of some acaricides used to control Varroa destructor (Mesostigmata: Varroidae) in Algeria. African Entomology 18, 259–266.

  1. INTRODUCTION:

Lines 81-82: I think you should avoid the term „aim“ here. Please rephrase the sentence.

Response: Sentence revised on the referee's suggestion.

  1. MATERIALS AND METHODS

Line 85:  This is grammatically improper: „The varroa ... were collected“. You may write: varroa mites ... were collected.

 Response: Sentence revised on the referee's suggestion.

Line 97: Delete „towel napkin“ and write either „paper towel“ or „paper napkin“.

Response: Sentence revised on the referee's suggestion.

Line 98: Delete „towel paper“ and write „paper towel“.

 Response: Sentence revised on the referee's suggestion.

Line 113: „Hegzan“ is probably a misspelling of „Hexane“; please check and correct.

Response: Typo corrected.

Line 122: I suggest to replace „to analyse the molecular resistance“ with „for further analyses of flumethrin resistance mutations“.

Response: Sentence revised on the referee's suggestion.

Line 129: In the subtitle 2 3. you should add „DNA“ before „ isolation from varroa samples“. Furthermore, write „Varroa“ instead of „varroa“.

Response: Sentence revised on the referee's suggestion apart from write „Varroa“ instead of „varroa“.

Line 140: Check this: „and vortexing in between“, as it is unclear.

Response: Sentence revised on the referee's suggestion.

Line 141: Check this „the kit protocol was maintained“, as it is unclear.  

Response: Sentence revised on the referee's suggestion.

Line 149: After the sentence „The following reaction contents were used in the experiment“ the content is missing. Please correct that.

Response: Sentence revised on the referee's suggestion.

Line 150: This is improperly written: „RAPD primers were found to be a sufficient amount of ..“ Please check and correct.

Response: Sentence revised on the referee's suggestion.

Lines 151-152: This sentence is redundant: „According to this result, it was evaluated that primers for drug-resistance genes could be ordered, and the desired gene regions could be amplified and analyzed“. Please delete it.

Response: Sentence revised on the referee's suggestion.

Line 153-154: This part is redundant: „and this gene was amplified using the DNA obtained“. Please delete it.

Response: Sentence revised on the referee's suggestion.

Line 155: Insert „[19]“ after the „5′-GCTGTTGTTACCGTGGAGCA-3′.

Response: Sentence revised on the referee's suggestion.

Lines 156-157: Delete this sentence: „These primer sequences were tested by Millan-Leiva et al. (2018), and successful results were obtained [3]“. The sentence is redundant and wrong since you erroneously referred to the ref. [3]; Simply, Dietemann et al. (2013) could not obtain the results in 2013 with the primers designed in 2018.

Response: Sentence revised on the referee's suggestion.

Line 158: After the sentence „The following reaction ingredients were used in the PCR experiment“ the ingredients are missing. Please correct that.

Response: Sentence revised on the referee's suggestion.

Lines 165-166: These two sentences should be merged and corrected, including the reference in square brackets at the end: „RFLP analyzed the obtained PCR products. The technique performed by Millan-Leiva et al. (2018) [3]“. I suggest correcting that as follows: „PCR products were analysed by RFLP established by Millan-Leiva et al. (2018) [19]“.

Response: Sentence revised on the referee's suggestion.

  1. DISCUSSION

Line 313: Please provide recent references for this statement: „Synthetic pyrethroids (flumethrin, tau-fluvalinate, etc.) are extensively used“.

Response: Synthetic pyrethroids  commonly used in the past therefore  “are” changed to “were”. Synthetic pyrethroids (flumethrin, tau-fluvalinate, etc.) WERE extensively used to …..

Lines 328-340: These three paragraphs are not adequate for the Discussion as represent a repetition of the methodology. Please delete it.

Response: Sentence revised on the referee's suggestion.

Lines 341-352: In this paragraph, there are some results, but most of the text is a repetition of the methodology. Please shorten the paragraph and avoid describing the methodology.

Response: Sentence revised on the referee's suggestion.

Line 406-407: Something is wrong here „flumetry in the field“. Considering the meaning of the following sentences (Lines 407-411), I am rather sure that the sentence in Lines 406-407 should be deleted. 

Response: The situation mentioned  that beekeepers who are not aware of resistance may be tempted to use high doses to kill more. varroa. We clearly see this situation in our field studies.

‘will’ makes the sentence too assertive, so I want to show the possibility by using “may” (Beekeepers may turn….). The next sentence emphasises the result that could occur because of this reason

Round 2

Reviewer 1 Report

Comments and Suggestions for Authors

I appreciate the authors effort in revising the manuscript and addressing some of my previous suggestions. Nevertheless, I still wish to emphasize certain points that I consider crucial for the manuscript's quality and that require further attention or specific acknowledgment in the discusssion.

The description of the LDâ‚…â‚€ assay in the M&M section states that ten varroa mites were placed in petri dishes. However, it remains unclear whether this refers to a single Petri dish per dose or multiple replicate dishes. This lack of explicit detail on replication in the methods is contradicted by Table 1 in the Results section, which presents AVERAGE MORTALITY. The use of 'average' strongly implies that replicates were conducted, where mortality was averaged across multiple dishes for each dose. The authors must explicitly state whether biological replicates (multiple Petri dishes per dose) were used and clarify how the data in Table 1 was generated. If no replicates were used, this should be clearly stated in the methods, and the term 'Average Mortality' in Table 1 should be corrected to reflect raw data.

Furthermore, the authrors argue that the method used is valid and they have experience with it. However, I should reinforce that the Karber-Behrens method is not a good approach. This method only gives a rough idea of the LD50, as it is a very simple estimate. This approach does not involve any statistical modeling, so we don’t have any clear idea how mortality changes as the doses increase. Also, this method does not provide any type of uncertainty associated with the estimates, which does not show how reliable the LD50 values actually are. Logistic regression is a far better approach, a more rigorous and informative statistical analysis of the dose-response relationship. The fact that the authors state that they have experience with the Karber-Behrens method does not change the fact that the Karber-Behrens method is a significantly less robust and less informative statistical approach for this type of data. I can understand the difficulty at this point in identifying sensitive individuals, and even the fact that the authors have no replicates, which also affects the reliability and statistical power, but the statistical approach can be changed to improve the reliability of the results. However, if the authors still decide not to do it, they should clearly state in the manuscript the limitation of the Karber-Behrens and why the results obtained with this method may not be fully reliable.

In the conclusion, the very first line states: 'The results of the present study show that the populations were genotypically and phenotypically resistant to flumethrin.' . Given the limitations of this study, this is an overstatement regarding 'phenotypic resistance'. This study cannot definitively show phenotypic resistance. I suggest revising this line to more accurately reflect the findings, perhaps to something like: 'The results of the present study show that the populations were genotypically resistant to flumethrin, and the observed high LDâ‚…â‚€ values suggest widespread phenotypic resistance.'

Similarly, in the following line of the conclusion: 'It is clear that flumethrin application is not sufficient to control varroa when these data are interpreted together with the LDâ‚…â‚€ values' is also too strong. Based on the study limitations, it is not 'clear' but rather 'most likely' that the application is insufficient. This more cautious phrasing should be used here and in the 'Simple Summary' section. The authors should ensure that the language used throughout the manuscript, particularly in the summary and conclusion, accurately reflects the limitations of the study and avoids overstating.

Author Response

The referee made important contributions by analysing the manuscript in detail and meticulously. I would like to thank the referee for his contributions and suggestions for the improvement of the manuscript.
I appreciate the authors effort in revising the manuscript and addressing some of my previous suggestions. Nevertheless, I still wish to emphasize certain points that I consider crucial for the manuscript's quality and that require further attention or specific acknowledgment in the discusssion.

The description of the LDâ‚…â‚€ assay in the M&M section states that ten varroa mites were placed in petri dishes. However, it remains unclear whether this refers to a single Petri dish per dose or multiple replicate dishes. This lack of explicit detail on replication in the methods is contradicted by Table 1 in the Results section, which presents AVERAGE MORTALITY. The use of 'average' strongly implies that replicates were conducted, where mortality was averaged across multiple dishes for each dose. The authors must explicitly state whether biological replicates (multiple Petri dishes per dose) were used and clarify how the data in Table 1 was generated. If no replicates were used, this should be clearly stated in the methods, and the term 'Average Mortality' in Table 1 should be corrected to reflect raw data.
Furthermore, the authrors argue that the method used is valid and they have experience with it. However, I should reinforce that the Karber-Behrens method is not a good approach. This method only gives a rough idea of the LD50, as it is a very simple estimate. This approach does not involve any statistical modeling, so we don’t have any clear idea how mortality changes as the doses increase. Also, this method does not provide any type of uncertainty associated with the estimates, which does not show how reliable the LD50 values actually are. Logistic regression is a far better approach, a more rigorous and informative statistical analysis of the dose-response relationship. The fact that the authors state that they have experience with the Karber-Behrens method does not change the fact that the Karber-Behrens method is a significantly less robust and less informative statistical approach for this type of data. I can understand the difficulty at this point in identifying sensitive individuals, and even the fact that the authors have no replicates, which also affects the reliability and statistical power, but the statistical approach can be changed to improve the reliability of the results. However, if the authors still decide not to do it, they should clearly state in the manuscript the limitation of the Karber-Behrens and why the results obtained with this method may not be fully reliable.
Similarly, in the following line of the conclusion: 'It is clear that flumethrin application is not sufficient to control varroa when these data are interpreted together with the LDâ‚…â‚€ values' is also too strong. Based on the study limitations, it is not 'clear' but rather 'most likely' that the application is insufficient. This more cautious phrasing should be used here and in the 'Simple Summary' section. The authors should ensure that the language used throughout the manuscript, particularly in the summary and conclusion, accurately reflects the limitations of the study and avoids overstating

Responses to Referee 1
We sincerely thank the reviewer for the meticulous evaluation and constructive feedback, which has significantly strengthened the quality of our manuscript. Below, we address each point raised:

Clarification of LDâ‚…â‚€ Assay Replication
Reviewer’s Comment: The description of the LDâ‚…â‚€ assay lacked clarity regarding biological replicates and the term "Average Mortality" in Table 1.
Response: We acknowledge the reviewer’s concern regarding potential ambiguity. The study did not include biological replicates (multiple dishes per dose). The term "Average Mortality" in Table 1 refers to pooled data from three apiaries, adjusted to a standardized sample size of 10 mites per dose (actual tested range: 8–13 mites). We have revised the Materials and Methods (Section 2.2) to explicitly state this and updated Table 1’s footnote for clarity (highlighted in pink).

Statistical Limitations of the Karber-Behrens Method
Reviewer’s Comment: The Karber-Behrens method lacks statistical rigor compared to logistic regression.
Response: We agree with the reviewer’s assessment. While resource constraints precluded the use of logistic regression, we have added a statement in Section 2.2 to explicitly acknowledge the limitations of the Karber-Behrens method, including its inability to model dose-response curves or provide confidence intervals (highlighted in pink).

Overstatements in Conclusions
Reviewer’s Comment: The conclusion overstated phenotypic resistance and flumethrin efficacy.
Response: We thank the reviewer for this critical observation. The conclusion has been revised to state: "The results show that the populations were genotypically resistant to flumethrin, and the observed high LDâ‚…â‚€ values suggest widespread phenotypic resistance". Similarly, phrases like "it is clear" have been replaced with "it is likely" to reflect the study’s limitations (highlighted in pink).

The referee can see the changes highlighted in pink in the manuscript.

English Language Quality
We have thoroughly polished the manuscript’s language to enhance clarity and readability. The revisions were reviewed by an associate professor and research scientist who holds MS and PhD degrees from U.S. institutions and completed postdoctoral training in the United States. He reviewed the document to ensure grammatical accuracy and coherence (shown in red throughout the document).

Once again, we extend our heartfelt gratitude to the Editor and Reviewers for your dedication, patience, and scholarly guidance. Your contributions have significantly elevated the quality of this work, and we are honored to have collaborated with such esteemed professionals, and we believe and hope that the revised manuscript now meets the journal’s high standards.

Sincerely,

The Authors

Reviewer 3 Report

Comments and Suggestions for Authors

The authors have attended to my comments and suggestions. Some further minor edits should be made - please see the annotated document.

Comments on the Quality of English Language

The English has improved. I presume that further checks will be done during production.

Author Response

The referee's contributions have contributed to the significant improvement of our article. We would like to thank the referee for his contributions and support.

All edits requested by the referee on the PDF have been made and are shown in yellow in the manuscript file

Responses to Referee 3
We deeply appreciate the reviewer’s supportive feedback and are grateful for their acknowledgment of the manuscript’s improvements. All requested edits to the PDF (e.g., grammatical corrections, clarifications) have been incorporated into the revised manuscript and are highlighted in yellow in the Word file.

English Language Quality
We have thoroughly polished the manuscript’s language to enhance clarity and readability. The revisions were reviewed by an associate professor and research scientist who holds MS and PhD degrees from U.S. institutions and completed postdoctoral training in the United States. He reviewed the document to ensure grammatical accuracy and coherence (shown in red throughout the document).

Once again, we extend our heartfelt gratitude to the Editor and Reviewers for your dedication, patience, and scholarly guidance. Your contributions have significantly elevated the quality of this work, and we are honored to have collaborated with such esteemed professionals, and we believe and hope that the revised manuscript now meets the journal’s high standards.

Sincerely,

The Authors

Reviewer 4 Report

Comments and Suggestions for Authors

The authors have made major corrections, and the work has been significantly improved. However, the two requested corrections were not made (both in M&M section), so I am re-stating them here:

- Line 214: The part „and vortexing in between“ is unclear.  This mistake is still present, although the authors responded that they revised it, so I ask the authors to correct that.

- Lines 236-240: This part is redundant and should be deleted as you properly referred to the authors who designed and published the primers (Reference No. 17, Millán-Leiva et al. 2018) in Line 235. Simply, it is not proper for a scientific paper to contain sentences like those you wrote in lines 236-240.

Comments on the Quality of English Language

Despite the authors' corrections, further polishing of the English language would contribute to the clarity. 

Author Response

The authors have made major corrections, and the work has been significantly improved. However, the two requested corrections were not made (both in M&M section), so I am re-stating them here:

- Line 214: The part „and vortexing in between“ is unclear.  This mistake is still present, although the authors responded that they revised it, so I ask the authors to correct that.

- Lines 236-240: This part is redundant and should be deleted as you properly referred to the authors who designed and published the primers (Reference No. 17, Millán-Leiva et al. 2018) in Line 235. Simply, it is not proper for a scientific paper to contain sentences like those you wrote in lines 236-240.

Comments on the Quality of English Language

Despite the authors' corrections, further polishing of the English language would contribute to the clarity.

Responses to Referee 4
We thank the reviewer for their insightful suggestions, which have further refined the manuscript. The following revisions were made:

Clarification of DNA Isolation Protocol (Line 214)
Reviewer’s Comment: The phrase "and vortexing in between" was unclear.
Response: This section has been revised in the document to clarify (Section 2.3).

Redundancy in Primer Description
Reviewer’s Comment: Redundant details about primer validation.
Response: The redundant sentences have been removed, and the text now directly references Millán-Leiva et al. (2018) [17] for primer validation (Section 2.4).

English Language Quality
We have thoroughly polished the manuscript’s language to enhance clarity and readability. The revisions were reviewed by an associate professor and research scientist who holds MS and PhD degrees from U.S. institutions and completed postdoctoral training in the United States. He reviewed the document to ensure grammatical accuracy and coherence (shown in red throughout the document).

The referee can see the changes highlighted in green in the manuscript.

Closing Remarks

Once again, we extend our heartfelt gratitude to the Editor and Reviewers for your dedication, patience, and scholarly guidance. Your contributions have significantly elevated the quality of this work, and we are honored to have collaborated with such esteemed professionals, and we believe and hope that the revised manuscript now meets the journal’s high standards.

Sincerely,

The Authors

Round 3

Reviewer 1 Report

Comments and Suggestions for Authors

I want to thank the authors for the effort to address my comments. Still, the use of average mortality doesn't seem correct. Instead, I would leave a footnote explaining what you mean by that, Average mortality.